# Calves Use an Automated Brush and a Hanging Rope When Pair-Housed

**DOI:** 10.3390/ani7110084

**Published:** 2017-11-09

**Authors:** Gosia Zobel, Heather W. Neave, Harold V. Henderson, James Webster

**Affiliations:** 1AgResearch Ltd., Ruakura Research Centre, Hamilton 3240, New Zealand; harold.henderson@agresearch.co.nz (H.V.H.); jim.webster@agresearch.co.nz (J.W.); 2Animal Welfare Program, Faculty of Land and Food Systems, University of British Columbia, 2357 Main Mall, Vancouver, BC V6T 1Z4, Canada; hwneave@gmail.com

**Keywords:** welfare, housing enrichment, grooming, dairy

## Abstract

**Simple Summary:**

Improving calf housing is growing in interest as standard management meets only the basic needs of calves. In an observational study, we found that young calves interacted with an automated brush and a rope when given the opportunity. There was less variation in how much calves preferred the brush to the rope. Some calves used the rope as much as or more than the brush. We suggest that rope is a feasible, cheap, and farmer-friendly environmental enrichment option for housed calves; nonetheless, provision of multiple enrichment options should be considered to encourage use and meet individual calf preferences.

**Abstract:**

Calf housing often only meets the basic needs of calves, but there is a growing interest in providing enrichments. This study described the behaviour of calves when they were given the opportunity to interact with two commonly available enrichment items. Female and male calves (approximately 11 days old) were pair-housed in 8 identical pens fitted with an automated brush and a hanging rope. Frequency and duration of behaviours were recorded on 3 separate days (from 12:00 until 08:00 the following day. Calves spent equal time using the brush and rope (27.1 min/day), but there was less variation in the use of the brush as opposed to the rope (coefficient of variation, CV: 23 vs. 78%, respectively). Calves had more frequent (94 bouts, CV: 24%) and shorter (17.8 s/bout, CV: 24%) brush use bouts compared to fewer (38 bouts, CV: 43%) and longer (38.3 s/bout, CV: 53%) rope use bouts. There was a diurnal pattern of use for both items. Frequency of play was similar to rope use, but total time playing was 8% of rope and brush use. Variability among calves suggested that individual preference existed; however, the social dynamics of the pair-housed environment were not measured and therefore could have influenced brush and rope use. Multiple enrichment items should be considered when designing improvements to calf housing.

## 1. Introduction

Calves are typically reared in groups in New Zealand, but the majority of calves internationally are housed individually (USA [1]; Europe [2]). Calf housing typically provides for the basic needs of the calf (e.g., shelter, bedding, and feeding), but commonly lacks environmental enrichments (e.g., interactive objects, manipulable substrates, and in the case of single-housed calves, social partners). Recent research has suggested pair housing may be a feasible option for farmers, and offers the benefits of group housing by providing some level of social contact [3,4,5]. Regardless of group size, the provision of enrichment items could improve calf welfare during commercial rearing. Enrichment items aim to promote natural behaviours, reduce negative and increase positive affect, and improve biological functioning of individuals living in captive environments [6]. The opportunities for enriching the housing environments of cattle have been recently reviewed and broadly categorised into social, occupational, physical, sensory, and nutritional enrichments [7]. Housing systems that provide more than one of these enrichment types permit animals to express their individual motivations and preferences [8]. For instance, the provision of social partners (e.g., social enrichment) increased play of calves [9], but it also increased cross-sucking [10], particularly if calves were limit-fed and not able to express their need to suckle (e.g., occupational enrichment, such as a non-nutritive teat [11]). This example suggests that multiple enrichment options are required to create calf housing that promotes good calf welfare. 

Brushes, both automated and fixed, have been growing in popularity on dairy farms [12,13] and offer both sensory and occupational enrichment. Adult dairy cows spend small portions of their day scratching their head and neck in indoor pens with and without brushes (7.4 vs. 1.2 ± 0.6 min/day, mean ± SED), respectively [14]), demonstrating motivation for scratching. In other species, it is suggested that early life exposure to enrichment items promotes interaction with these items later in life [15,16]; given that the adult cows in the former study likely did not have brush use opportunities as calves, this may explain the relatively low use reported.

Cows may not be able to satisfy a motivation to scratch their full body without the provision of an object that allows for flexible interactions. Indeed, when able to access automated brushes (e.g., spinning brushes that cows could position themselves under) cows spent about 20% of their brush use time focused on their back and tail [14]. There are a number of factors that impact brush use, such as competition [14,17], temperature and feed location [18], and health status [19]. Reduced novelty can also decrease use (e.g., beginning with 2.1 ± 4.6 min/day and reducing to 1.3 ± 2.2 min/day after 4 weeks of exposure, mean ± SD [19]). Unpublished work has suggested that brush use can be determined by the type of brush (e.g., fixed or mechanical) [20], lack of other stimuli (e.g., individual housing) [21], and age (e.g., group-housed calves between 40 and 98 days of age used brushes between 4.5 and 9.0 min/day [22]). To the best of our knowledge, the use of brushes by young, milk-fed calves has not been assessed; it is therefore unclear how pair-housed calves that are less than 2 weeks of age and naïve to brushes will interact with them. Calves naturally receive frequent grooming from their dam [23], that is removed under commercial conditions and thus there may be a behavioural deficit. We suggest that commercially reared calves may seek out grooming opportunities even at a young age. 

While automated brushes provide sensory and occupational stimulation in older animals, it is possible that young, pre-weaned calves will benefit from the opportunity to satisfy a need to lick and orally manipulate an item. Beef cattle interact with lengths of manila (e.g., hemp-based) rope [24,25], but rope use was affected by temperature (e.g., decreased use in cold weather), and group size. Within a large feedlot, 50% of the cattle did not make contact with the rope in the first few hours of exposure, and some took up to 2 weeks from initial introduction to reach peak use [25]; however, total time spent using this rope was not recorded. Given that weaned feedlot cattle (approximately 8 months of age) interact with rope, milk-fed dairy calves with a strong suckling motivation may also seek oral manipulation, since this behaviour is greatly restricted in many calf rearing systems. In addition to oral manipulation, contact with the rope over the rest of the body may discourage fly activity; this behaviour has been noted in elephants that frequently use vegetation to displace insects [26]. The potential benefits of providing rope have not been documented in cattle.

The aim of this study was to document how and when dairy calves use an automated brush and a hanging manila rope. We also recorded play behaviour (including locomotive, social, and ground play) as a comparison measure with brush and rope use in order to understand how, and how often, calves interact with different aspects of their environment. We hypothesized that calves would readily use both enrichment items, but that brush use would be more consistent over time and that rope use would increase over time. 

## 2. Materials and Methods

This study was conducted from March to May 2016 at the AgResearch Tokanui Dairy Research Facility, Te Awamutu, Waikato, New Zealand and was approved by the Ruakura Animal Ethics Committee (Application #13822) (Hamilton, New Zealand) under the New Zealand Animal Welfare Act 1999 and by the University of British Columbia Animal Care Committee (#A14-0245) (Vancouver, BC, Canada).

Eight pairs of calves (KiwiCross) were housed in identical pens. Seven pairs were mixed sex (i.e., one male and one female), and one pair was female. Within these pairs, one calf per pair was randomly selected for colour marking (Tell Tail paint, FIL NZ Ltd, Mount Maunganui, New Zealand) to distinguish between animals during video observation. From each pair, a focal calf (4 females and 4 males) was selected to ensure that sex was balanced (the first 4 pairs had a focal calf randomly selected and the sex of these dictated the focal calf selected in the other 4 pairs). At enrolment, focal calves weighed 54.0 ± 3.3 kg (mean ± SD). Exact age was available for only the females (11.5 ± 1.3 days old, mean ± SD); however, it was known that the males were born within the same week as the females. Pairs were housed in eight plywood enclosed pens (Figure 1) that provided 5 m^2^ per calf, that were bedded with wood shavings and contained a wall-mounted automated brush (mini swinging brush MSB, DeLaval, Sweden), a 1 m long rope attached to the pen wall opposite the brush (hemp-based manila rope, 28 mm, 3 strand, Action Outdoors Ltd, Auckland, New Zealand), and a small pile of straw, replenished every second day.

Calves were fed 4 L of whole milk from a 5-nipple milk feeder twice daily at 07:00 and 15:00; milk feeders were left in the pen for 30 min to ensure the full milk meal was consumed. Each milk meal also contained 12.5 g of Exagen (Professional Veterinary Distributors Ltd, Auckland, New Zealand), a product that is recommended to reduce the risk of cryptosporidial scours. Calves had access to ad libitum hay, concentrate (SealesWinslow Calf Pro, Mount Maunganui, New Zealand), and water from wall-mounted drinkers that were cleaned and replenished daily. Fly spray (approximately 5 mL/pen, Permoxin Insecticidal Spray, Dermcare-Vet Pty Ltd., Springwood, Australia) was applied to the pen walls and coat of each calf at 11:30 every second day.

Health of the calves was monitored visually twice daily at feeding, and formal health checks were performed at the time of enrolment according to an established protocol [3]. Any calves that were suspected to be ill were inspected, treated by a veterinarian, and were not utilised in this study.

Video cameras recorded the calves’ behaviour (DS-2CD2432F-I(W), HikVision, Hangzhou, China) and were connected to HikVision NVRs and software (DS-7732N1-14/16P and DS-771NI-SP, HikVision, Hangzhou, China). Calves were given 24 h of acclimation time (d0) to the pen prior to the beginning of observation. Frequency and duration of behaviours on d1, d2, and d6 were coded using continuous recording; observation periods on each day were comprised of 20 h, from 12:00 to 08:00 the following day. The remaining 4 h were not included due to management factors and disruptions in the housing facility. Behaviours recorded each day were brush use, rope use, and play behaviour (Table 1).

All summaries and statistical analyses were performed with SAS (version 9.4; SAS Inst. Inc., Cary, NC, USA). One observer scored all behaviours after establishing high inter-observer (brush use, κ_W_ = 0.97; rope use, κ_W_ = 0.86; play behaviour, κ_W_ = 0.91) and intra-observer (brush use, κ_W_ = 0.89; rope use, κ_W_ = 0.96; play, κ_W_ = 0.81) reliabilities based on 8 h of video of one calf. One male calf was removed from the dataset due to faults in the video file. For the remaining calves (*n* = 7), hourly and daily bouts (no), bout duration (s/bout), and total duration (min) of each behaviour (brush use, rope use, and play behaviour) were calculated. Mean and coefficient of variation (CV, %) were calculated across days. The effect of day on brush and rope use was analysed using a mixed model with day and sex as fixed effects and calf as a random effect. Pearson correlations were calculated between total duration and bouts for play behaviour and brush use, and for play behaviour and rope use, using the mean of the 3 days for each calf. Diurnal brush and rope use patterns were analysed in separate hourly mixed models with day and sex as fixed effects and calf as a random effect. To determine whether calves used the two resources similarly, a brush to rope use ratio was calculated for each calf for each of the 3 days. The ratios were then log_10_ transformed, and analysed in a mixed model with day and sex as fixed effects and calf as a random effect. Final models included only significant effects. Significance was set at *p* ≤ 0.05 and tendency at 0.05 < *p* ≤ 0.1. 

## 3. Results

### 3.1. Brush Use, Rope Use, and Play Behaviour

Calves spent 85% of the time using the brush on their head and neck area, with the remainder of time split equally between the middle of the body and the hind quarters (Table 2). Calves interacted with the brush frequently and for short periods of time; there were nearly 3 times as many brush bouts, and these were less than half the duration, as the rope bouts (Table 2). However, the total time spent interacting with each item was nearly identical (27.1 min/day). Calves were similar to one another in their brush use, but rope use was highly variable among calves. Day did not affect rope bouts and bout duration or any of the brush use measures (*p* > 0.1); however, total duration for rope use was lower on d1 compared to d2 and d6 (15.8 vs. 29.8 and 34.8 ± 7.4 min/day, mean ± SED; *p* = 0.06). Sex did not affect any of the brush or rope use measures (*p* > 0.1). Play was observed for 2.1 min/day, and play bouts and total duration were highly variable among calves. There was a tendency for a correlation between total duration of brush use and play behaviour (*r* = 0.73, *p* = 0.06), but there was no relationship for bouts. There was no correlation between rope use and play behaviour for either total duration or bouts. 

### 3.2. Hourly Distribution of Brush Use and Rope Use

Brush use increased around feeding time (15:00 and 07:00). Following the afternoon feeding, the increased use continued for 4 h (Figure 2a). A similar hourly distribution occurred for rope use, but there was more variability among hours (Figure 2b). Bouts followed a similar pattern to time, but the number of brush bouts (Figure 3a) was more pronounced than the number of rope bouts (Figure 3b). The hour of morning feeding (07:00) had the longest total time spent using the brush across the 3 days; one calf interacted with the brush for 12.6 min/h (d2), and another interacted with the rope for 16.0 min/h (d2). A similar pattern was noted 2 h after afternoon feeding (17:00), when one calf on d6 had total times of 12.4 min/h and 15.1 min/h of brush and rope use, respectively. Due to management and disruptions in the facility, it was not possible to monitor brush and rope use after morning feeding (08:00–12:00).

### 3.3. Brush to Rope Use Ratio

Calves tended to differ in their ratio of brush to rope use (Figure 4). Most calves had a high brush to rope use ratio (i.e., they used the brush more than the rope); however, one female calf used the rope more than five times as much as the brush. There was no effect of day (F_2,12_ = 1.1, *p* > 0.1), demonstrating that, over a span of 6 days, the calves were consistent in how much they interacted with the brush compared to rope. Sex did not affect brush to rope use ratio (F_1,5_ = 0.04, *p* > 0.1).

## 4. Discussion

We described the daily and hourly use of an automated brush and a manila rope by young (average 11 days old) dairy calves housed in pairs. After one day of acclimation the calves interacted with both items. On average, over the 6 days period, calves spent nearly 30 min/day interacting with the brush (daily observations were 20 h). Previously reported times range from as low as 1.3 min/day for adult cows after 4 weeks of exposure [19] to 9 min/day for 40–71 day old calves [22] housed in competitive, group environments. It is possible that the pair housing and large space allowance [28] in the current study reduced competition between the calves for the two resources, which may account for higher use compared to other studies. However, competition and social facilitation between the focal calf and its penmate was not monitored within the current trial. We suggest that stocking density may impact the use of, and displacement from, brushes; it would be valuable to determine the use of these items by single-housed calves where there is no social facilitation, but arguably the need for enrichment opportunities is greater. More broadly, measuring the motivation of individuals to access these items (as has been used to measure motivation of cattle to access pasture or food [29]) would provide valuable insight into the importance of a brush or rope for young calves, even if the items are not used often.

Interestingly, the time spent with the rope was nearly identical to the brush. We anticipated that rope use would increase over the course of the 6 day observation period; indeed calves used the rope less on the first day compared to the other days of observation. However, the way in which calves interacted with the items and individual calf differences in the use of each item were the key factors distinguishing the two resources. Calves interacted with the brush in many, short bouts, while bouts of rope use were fewer and more prolonged. This suggests that the calves valued these resources in different ways. For instance, the brush had an interactive component (e.g., it would swing/swivel in response to pressure from the calf, and would also spin automatically for 10 s following contact); thus, the movement of the brush may have promoted on-going visual and physical interaction after a calf finished a bout. Conversely, the rope was orally pliable and promoted chewing and sucking behaviours [24,25], which could not have been achieved with the brush. It is important to note that the variability in use of both items may have been due to the pair housing environment that could have promoted social facilitation (e.g., one calf interacting with an object triggers the other calf to interact as well) or prevented resource use (e.g., one calf was dominant over a preferred resource). 

Individual calves were more similar to one another in their brush use bouts, bout duration, and total duration than they were in their rope use, suggesting that some individuals may have preferred one resource over the other. For instance, we found that 4 of the 7 calves had a high brush to rope use ratio [30]. However, two calves did not favour one item over the other (e.g., used both rope and brush the same amount) and one calf spent more than five times as much time with the rope as the brush. While it is beyond the scope of this study to determine causality of use, we encourage further work using a larger number of calves to examine whether individual variability is due to individual preference, social factors such as social facilitation, or dominant–subordinate relationships affecting access to the items. Variability may also be related to novelty of the objects; while calves used the items for a total of nearly 1 h for every 20 h of observation in the first week of exposure, the number, or novelty, of the items may be important factors maintaining use among individuals over time [31]. 

There was a distinct diurnal pattern of brush and rope use. In the hour prior to afternoon feeding (14:00), rope use increased, temporarily reduced around feeding (15:00–16:00), and then largely increased again following feeding (17:00–19:00). Brush use followed a similar timeline, except there was no decrease around feeding. The similarity in resource use suggests that the interaction with these items is due to a general diurnal increase in activity, rather than a desire to use one item over the other. Indeed, hourly use for individual calves was highest for both the brush and the rope during the hour of morning feeding (e.g., two individuals spent 20–27% of their time interacting with the brush and rope, respectively). Conversely, a pilot study that examined brush use in calves only found an evening peak in this behaviour [22]. However, those calves were significantly older and weaned from milk, whereas the calves in the current study had a morning milk feeding that may have promoted activity at this time point.

These patterns of brush and rope use may be related to redirection of a motivation to engage in a particular behaviour that is unable to be satisfied in the environment. Increased rope use around feeding time may be a redirection of a motivation to suckle through oral manipulation of the rope. This behaviour may initially be a response to increasing hunger before feeding and then later a response to a desire to continue suckling after available milk has been consumed [32]. Calves are particularly motivated to perform non-nutritive sucking (i.e., sucking on a dry teat without milk delivery) or cross-sucking (i.e., sucking on another calf) when fed limited amounts of milk and following a milk meal [11,33]. Although calves in our study were offered milk in 4 L volumes twice per day and feeders were removed after 30 min, this may have stimulated non-nutritive sucking before and after feedings that was directed at the rope. We were unable to determine if calves were sucking on the rope, and cross-sucking on the other calf was not specifically recorded. Further investigation is needed to understand the motivations behind rope use; for instance, if calves reduce rope manipulation when also offered a dry teat, this would suggest rope use is driven by a suckling motivation, but if rope use is maintained, this would indicate that the rope has value independent of satisfying an oral manipulation desire. 

Brush use has also been suggested to be a redirected behaviour in adult dairy cattle. Brush use was increased in the weeks after calving, suggesting an increased motivation for tactile stimulation at this time [34]. Because calves are removed from their mother immediately after birth, the authors speculated that brush use may be a redirected behaviour resulting from a motivation to engage in contact with their calf [34]. Furthermore, work has shown that cows did not use a brush in the presence of their calf, but did so after the calf was removed [35]. A similar explanation can be applied to the use of brushes by young dairy calves who are unable to achieve maternal contact and may seek stimulation from the rotating brush that may mimic maternal grooming. 

Calves in the present study focused the majority (85%) of their interactions with the brush on their head and neck area. This was expected as it has been previously reported for older cows [14,36]. It is also possible that the calves were seeking relief from itchiness associated with flies. While fly cover was not quantified, a large presence necessitated the use of fly spray every other day, which may have been bothersome to the calves.

Play behaviour was observed for a few minutes each day, and was highly variable among calves for both bouts and total duration. This is corroborated by previous work suggesting that play behaviour occurs during small portions of the day [27,37]. Play behaviour may be related to the age of the calves. Pair-housed calves played more at 2 weeks than at 6 weeks of age [38]; however, it is worthwhile noting that this reduction in play could have been an artefact of the calves’ response to the removal of milk. Nonetheless, play is suggested to be an important indicator of good welfare [39] and is thought to be a pleasurable experience [40]. The positive relationship between duration of play and brush use in the current study suggests that brush use may also be pleasurable.

While we noticed an increase in interactions with the rope, this trial only focused on a 6 day period, and therefore did not capture any change in longer term enrichment use. In addition, brush and rope use may be related to the age of the calf and the amount of milk that is offered. For instance, it has been shown that brush use by calves decreases with age [16]; a similar decrease in use was noted for adult cows [19], which could be a function of reduced novelty. Brush and rope use may also have been influenced by the social environment, particularly if the focal calf was more dominant or subordinate at these resources compared to their penmate. Cattle are known to exhibit dominance over specific resources [17], but little is known about how the social relationships among young calves or group size impacts use of particular resources. Importantly, the functional relevance of brush and rope use is still unclear; these additions to calf housing may serve as an object of redirected behaviours associated with maternal grooming or motivation to suck, or they may serve specific functions for scratching or playing, or offer pleasurable properties that are yet unknown. 

## 5. Conclusions

In conclusion, young calves used both an automated brush and a hanging manila rope for nearly 30 min per item per day (over a 20 h observation period) when in pair housing. There was a clear diurnal pattern of use, with both items used heavily following afternoon feeding, in early evening, and before morning feeding. The variability in brush to rope use between individual calves suggests that there are individual preferences for different enrichment items, but other factors such as social facilitation and hierarchy likely also affect use of these items. Although rope is inexpensive for farmers to install in calf pens, both brushes and ropes should be provided in calf housing to allow calves to exercise their preferences.

## Figures and Tables

**Figure 1 animals-07-00084-f001:**
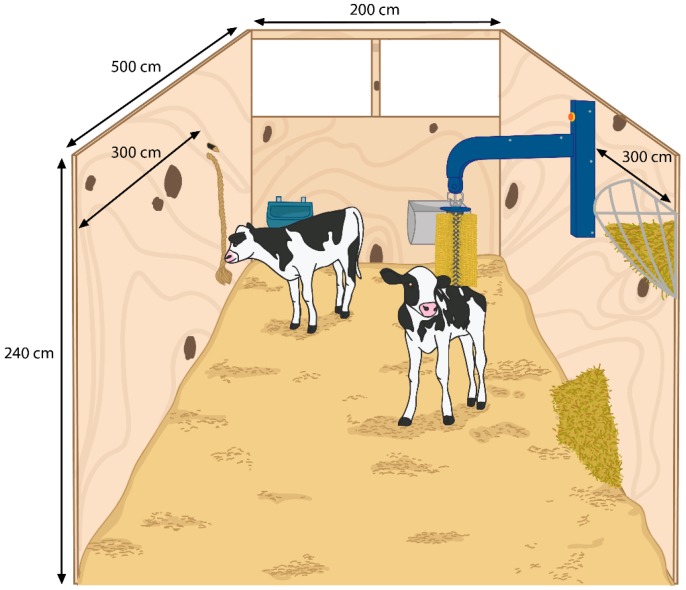
Pen layout for 8 identical pens containing an automated brush and a 1 m length of manila rope. Water, concentrate, and hay were provided ad libitum. Pens were bedded with wood shavings and housed 2 calves. One calf per pen was selected as the focal animal for observation.

**Figure 2 animals-07-00084-f002:**
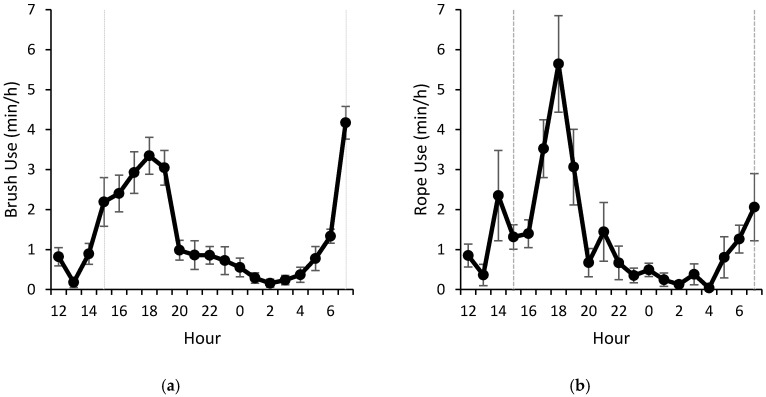
Mean (±SEM) time spent using (**a**) brush and (**b**) rope for 7 calves averaged for 3 days (d1, d3, and d6). Observations spanned from 12:00 to 08:00 the following day. Calves were fed 4 L of whole milk from a 5-nipple milk feeder twice daily at 07:00 and 15:00 (dotted line); feeders were left in the pen for 30 min to ensure the full milk meal was consumed.

**Figure 3 animals-07-00084-f003:**
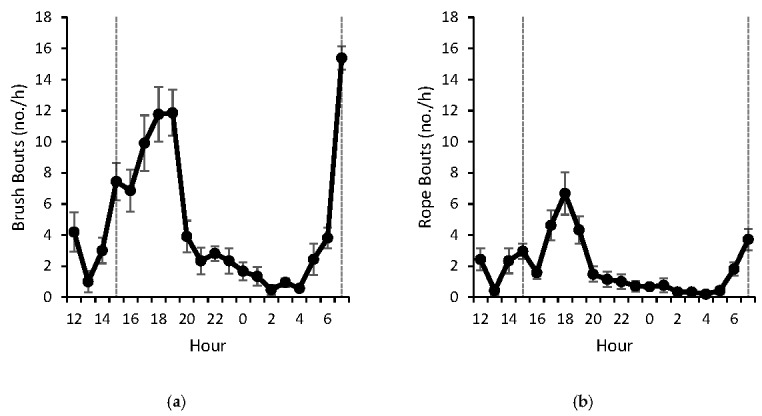
Mean (±SEM) number of (**a**) brush and (**b**) rope bouts for 7 calves averaged for 3 days (d1, d3, and d6). Observations spanned from 12:00 to 08:00 the following day. Calves were fed 4 L of whole milk from a 5-nipple milk feeder twice daily at 07:00 and 15:00 (dotted line); feeders were left in the pen for 30 min to ensure the full milk meal was consumed.

**Figure 4 animals-07-00084-f004:**
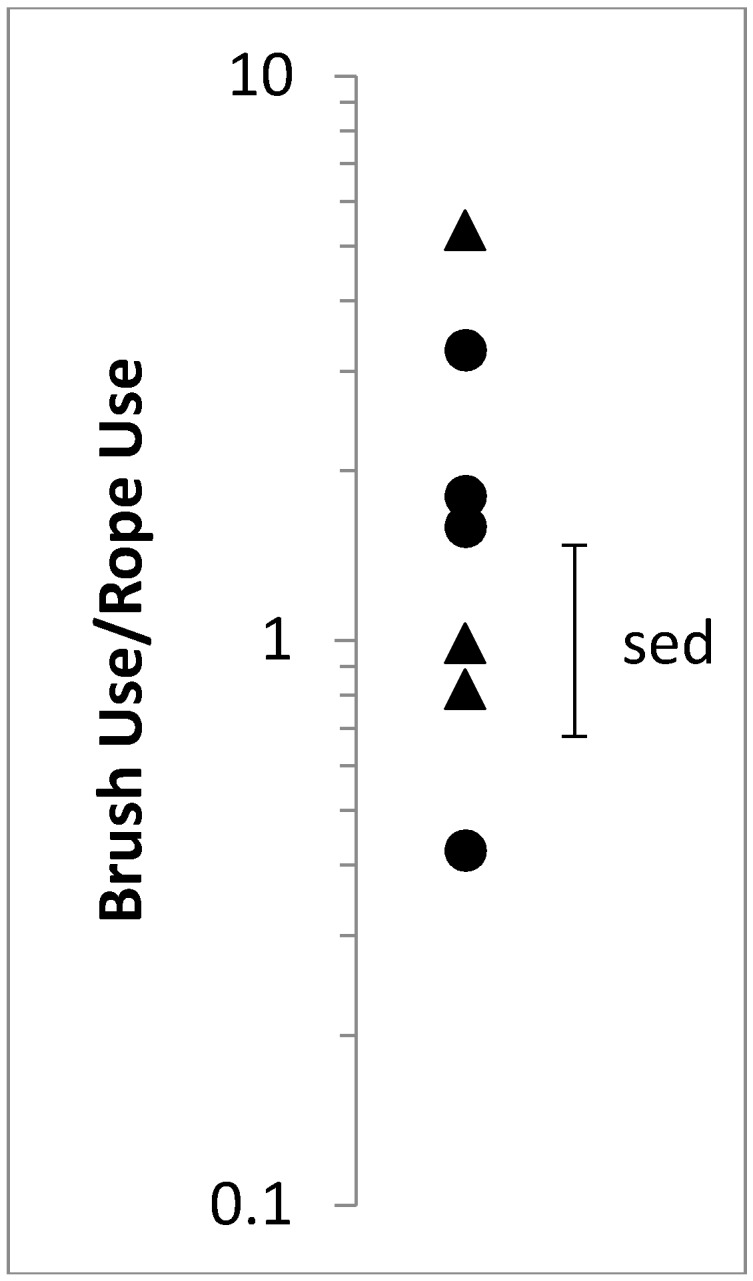
Brush to rope use ratio presented as a mean for 3 days (d1, d3, d6). Observations spanned from 12:00 to 08:00 the following day. The mean of each calf is represented by ● (female) or ▲ (male). Values are back-transformed and presented on a log transformed scale to maintain SED.

**Table 1 animals-07-00084-t001:** Ethogram of calf interactions with the brush and rope, and play behaviour.

Variable ^1^	Description
Brush interaction	
Head	Head region of body (cranial to withers/front legs) in contact with mechanical brush
Mid	Mid region of body (torso from withers to hips; between front legs and back legs) in contact with mechanical brush
Hind	Rear of body (from hips to rear end; behind back legs) in contact with mechanical brush
Rope interaction	Any part of body is moved against rope (scratching/grooming/chewing); does not include calf standing still while in contact with rope
Play behaviour ^2^	Locomotive play (gallop, jumping, leaping, bucking/kicking, turning, movement upwards and sideways), social play (butting another calf, mock fighting), and ground play (rubbing the head, throat or neck in the bedding while kneeing down on both forelegs) ^2^

^1^ Must be at least 2 s between bouts of behaviours to be scored as two separate bouts. ^2^ See [27] for specific definitions of each play behaviour.

**Table 2 animals-07-00084-t002:** Descriptive statistics for brush use, rope use, and play behaviour of 7 calves averaged for 3 days each from 12:00 to 08:00 the following day.

Variable	Bouts (no./day)	CV ^2^ (%)	Bout Duration (s/bout)	CV ^2^ (%)	Total Duration (min/day)	CV ^2^ (%)	Proportion of Days (%)
Brush interaction							
Total calf ^1^	94	24	17.8	24	27.1	23	2.3
Head	68	22	20.6	20	23.1	25	1.9
Mid	13	56	11.2	60	2.4	71	0.2
Hind	13	58	7.2	79	1.6	82	0.1
Rope interaction	38	43	38.3	53	27.1	78	2.3
Play behaviour	37	62	3.4	30	2.1	66	0.2

^1^ Sum of all instances the calf used the brush on its head, middle of its body, and hind quarters. ^2^ Coefficient of variation (%) provides a measure of the dispersion of the means of each calf over the 3 days (the higher the %, the higher the variance between the individual calves).

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
