# Peer review of "Calves Use an Automated Brush and a Hanging Rope When Pair-Housed"

_animals, 2017, doi:10.3390/ani7110084_

Reviewer 1 Report

General comments: an interesting, simple and clear study, easy to understand but with great potential to suggest a nice improvement for calf welfare. Overall, consider a few minor grammatical changes, especially in the introduction. Only other minor suggestions would be to integrate some more literature references into the intro and discussion to help better contextualize the paper in existing research/animal welfare approaches.

Simple summary

I find the introductory sentence of the simple summary to be somewhat odd- worth taking another look?

Abstract:

L20: what do you mean the behaviours were recorded on 3 d? they were recorded every 3 days?

L26: time playing was one tenth the time of what?

Introduction

The authors may consider providing some more background info on environmental enrichment generally, what it seeks to do, what some of the challenges are to implementation, etc. etc. See Newberry 1995, for example, and other related articles. As of now the authors assume the reader understands why enrichment is important for welfare.

L32: Grammar in first sentence doesn’t make sense.

L58: Might it also be possible that the reported low use of brushes by adult cattle may indeed relate to the lack of provision in young calves? Would be interesting for authors to expand discussion of importance of early provision of enrichment objects and implications of such

L59: how costly?

Materials/methods

LOVE the illustration!

All descriptions clear.

Results

L148 Presumably when article goes to press, the table will not be split across pages…

Figure 2a: You likely address this in discussion, but am I interpreting correctly that brush bouts increased AFTER afternoon feeding, but BEFORE morning feeding? If so, why do you suppose this is? If you don’t already discuss this, would be interested in hearing author perspectives on this…

Discussion

The discussion could benefit with a bit more reference to other studies- for example in promotion of important motivated behaviours like chewing and sucking, discussion of individual preferences shown to enrichment, etc.

L228: Did you explain already why you did not record instances of cross-sucking? Why didn’t you? I realize it would have been more interesting to collect this if you had also been comparing enrichment v. non-enriched treatments…room for future studies!

Author Response

Thank you to the reviewer for their comments!

We have addressed each point individually and because we have included some graphs have included the response as a PDF.

Reviewer 2 Report

This study provides data on circadian patterns of usage for two potential environmental enrichment for pair-housed crossbred calves. It provides useful information about patterns of use, and some insight into individual preferences (although the latter was slightly confusing - see detailed comments below). Data collection within the period studied is thorough and the consideration of observer reliability is appreciated. However, the observation period was quite short, so the applicability in terms of long-term preferences is limited, and conclusions can only be drawn about calves in this particular age range.

Detailed comments by line number

49-50 - Does the object really have to be "interactive" e.g. rotating to allow them to scratch body parts they can't reach themselves? Why?

56 - I found it unclear whether this was variable on an individual basis or between studies and perhaps age groups or breeds. If the former, the values given should include measures of variation. If the latter, maybe worth elaborating on possible reasons so you can discuss how this study fits?

69 - Can calves really scratch themselves using a rope?

Table 1 - Should read "mock fighting", not "mocking".

131-132 - fixed effects should be plural

143-144 - Where is the data indicating that they are similar to one another in duration compared to rope use? No stats are given here and section 3.3 then states they were not all similar.

173 - "lower brush to rope use ratio" - lower than what? Do you mean they used the brush less than the rope?

175 - Based on the section this is in, I assume this means they were consistent in their preference for brush vs. rope, rather than actual usage of each object, but this should be clarified and it would be nice to see actual usage data for each over days.

261 - Wouldn't ratio of use be the clearer indicator of whether there is individual difference in preference? 

265-266 This seems a bit out of place in the conclusions, at least not without more explanation. This study didn't show anything about the importance of choice directly. If the implication is that perhaps farms should give both items at once to allow expression of the individual preferences, perhaps this could be better linked to the preceding statement; this is only one aspect of the importance choice is believed to have.

Author Response

The authors acknowledge the reviewers attention to detail and the request to review the statistical methodology in particular. We have now clarified the methodology and how the results are presented.

We have attended to each comment and since we have included a graph, the response is attached as a PDF.

Thank you!

Reviewer 3 Report

Manuscript: Enrichment options for pair-housed dairy calves

General comments:

This research examines how (frequency, duration etc.) dairy calves housed in pairs interact with a rope and/or an electronic brush. In doing so, this research provides further detail on the behaviour and requirements of young dairy calves, and I think it is important that we continue gaining knowledge in this area. In general, the research is well written and presented. Therefore, I think this research is worthy of publication.

However, there are a number of limitations to the research and I think the manuscript can be strengthened by a better consideration of what is enrichment, and an acknowledgement of some of these limitations. Below I have notes my more significant and general concerns, before discussing more specific comments/recommendations.

More significant comments:

1.      Just because an animal is provided with something (rope, brush “toy”) does not necessarily mean it is enriching or results in an improvement in animal welfare. Ruth Newberry has written a good paper addressing this that may be useful to this paper (Newberry, 1995. Applied Animal Behaviour Science, 44 (1-2): 229-243). In light of this, I think the term “enrichment” needs to be removed from the title, and for the title to more specifically refer to rope and brush usage. In particular, both the bush and the rope were not used very frequently (total duration 27.1 mins per 20 h) – so a better consideration on whether these are actually “enriching” and whether they result in improved animal welfare is required. This could be better considered. I note that the author has done this a little (e.g., lines 236-238), but use of the rope could just be a sign of hunger and thus indicative of poor welfare, rather than allowing expression of a highly motivated behaviour which may result in improved welfare. The same needs to be assessed for the brush - for example, does it mimic maternal grooming? Indeed, the fact that play behaviour was so infrequent and variable between animals could suggest low welfare, although perhaps calves that young do not engage in play behaviour (hiding phase of development)?

2.      I don’t really understand how play is used as a comparison for time interacting with enrichment. Are you saying the play per se is an enriching behaviour? I suppose this may be true, but it is also an indicator of welfare (animals with better welfare are more inclined to play, or the environment promotes play behaviour which increases welfare), which you mention (briefly) in the discussion. Also, play behaviour increases with time and 11 d old calves may still be “hiders”, meaning play may not be expected to occur much. So, can you please expand on how you expect a comparison to play behaviour to provide you with details on the relevance of the brush/rope? You mention cross-suckling in the introduction, but I’m guessing it is not something you looked at? I would find it more relevant to assess whether the calves that utilised the rope spent more time playing and less time cross-suckling (if recorded), and infer from the results whether the rope has functional significance.

3.      Social effects. If one calf is occupying the brush/rope this prevents the other from getting access. This raises questions of whether variation in brush/rope ratio is attributed to true “preference” or rather just an artefact of social dominance relationships. This requires further discussion.

4.      Statistics. Overall, you have low animal numbers (n=7), so all of you discussion needs to be moderated by this. The statistical analyses is a bit confusing, partly because of the use of term “observation period”, which it seems can refer to day (or 20 h per 24 h) or hour within day. Further, is there a reason you had calf as a fixed effect and not a random effect? It may also have important to control for the penmate, or pen location. Whether this is done in effect by controlling for calf (as there was only one focal calf per pen) or not I don’t know, but maybe you could explain or consult a statistician to make sure.  Sex should probably be included as a random effect also. You also specify hour as a fixed effect, but there is no mention of day even though you report on day in the results. For both day and hour, you are performing repeated observations of the same animals over time, so the model should account for this with repeated measures and this is also not specified (I don’t know if it wasn’t done or if it was done and you haven’t specified). If it wasn’t done, then each time point really needs to be analysed separately and this would also need to be made clear.

More specific/less significant comments:

1.      Title

a.      As previously discussed, I think the title should be changed to more accurately reflect the comparison in this study i.e., brush vs. rope.

2.      Simple summary

a.      Sentence lines 11-12 is difficult to read – do you mean “While brushes were used consistently by all calves” rather than “While brushes were used MORE consistently by all calves”? Or do you mean there was less variation in how often calves used the brush?

3.      Abstract

a.      Line 18: What sex were the calves?

b.      Line 19: “outfitted” change to “fitted”

4.      Introduction

a.      Lines 32-34: I don’t think this addition detail on NZ practises is needed, especially considering this is an international journal and (as you say)pair housed calves is more common and relevant internationally.

b.      Line 36: “Environmental enrichments” – this is a very broad term can you provide some examples (social interactions, bedding, manipulable substrates (e.g., bedding)).

c.      Line 43-44: I can see how preventing cross-suckling could improve welfare of the calf being cross-suckled, but, if the cross-suckling calf is still hungry is its welfare improved by non-nutritional suckling? I suggest removing this, unless you have a reference that could back it up.

d.      Lines 57-58: Has anyone given a brush to 11 d old calves? Are they motivated to use a brush? This comes back to my earlier point about the functional relevance of the enrichment.

e.      Lines 59-60: In adult cattle it seems this statement is true, but what about young animals? Is it functionally relevant to young calves?

f.       Lines 72-73: I don’t really understand how play is used as a comparison for time interacting with enrichment, as discussed in the more significant comments.

5.      Materials and methods

a.      Lines 83-84: I don’t think you need to refer (here or elsewhere) to the other study that was being conducted on the calves. It’s not really relevant.

b.      Line 85: On what basis were animal selected? Can you clarify whether they were in single sex groups.

c.      Line 99: Can you confirm that there were enough nipples for both calves in the pair to feed simultaneously?

d.      Line 106: as per comment 5a, remove …“following management associated with the larger study”…

e.      Line 115: replace “because” with “due to”. Remove …”associated with the larger study in which the calves were enrolled”. 

f.       Line 117-120: This is good, but I think it belongs in the statistics section.

g.      Line 126: was the calf removed a male or a female?

6.      Statistical analyses

a.      Line 127: I get confused by the use of the word “period”. Is this just each 20 h? I think if you define that this is what you are calling a “day” you could just use per day, it would be easier to understand especially later on when you go into detail how the brush/rope were used over hours within days.

b.      Line 130: Be more explicit here – across both observation days and hours within days?

c.      Line 130-131: Previously you used the term “observation period” to describe hour and day periods, but here you have used the term observation period and then specific hour as a fixed effect. Does this analyses refer to only the hours within days? Or also to days? Also, for the analyses pre day, did you analyse each day separately? If not, the analyses would be a repeated measures (and is for hour too!), and so the model needed to account for repeated observations of the same animals over time.

d.      Line 132: Surely there were other random variables to control for? What about sex effects? And penmate? Pen location (i.e,. sometimes being in areas of high human activity can disrupt or encourage particular behaviours)? Perhaps calf should be nested within pen as a random effect – is there a reason you put calf as a fixed effect?

7.      Results

a.      Line 145-146: But what age to calves typically start playing? They may be too young for play to be frequent of bouts of play to be long. There may also be sex effects, but you haven’t included sex in your model.

b.      Line 155 – use of brush temporal or related to feed? i.e., would a calf normally be groomed by the dam post-feed? Alternatively increased activity at sunset could be associated with general increased activity, as dairy have their longest/largest meal at sunset? This needs to be discussed.

c.      Line 159-160: One calf maximum use 12.6 min/hr – what does “one calf” mean? Is this the maximum use by a single calf? Or do you mean of the pair of calves one of them used the brush/rope the maximum of 12.6? Were these high(er) usages outliers? With low animal numbers an outlier could increase the mean substantially.

d.      Remove “associated with the larger study in which the calves were enrolled”

e.      Line 174: You report on effects of day but this is not included in the statistical analyses.

8.      Discussion

a.      186-188: Also possible that calves are too young for these enrichment sources to have a functional significance. Need to also consider whether such low usage actually means anything to the animal.

b.      Line 194: replace “develop” with “increase”?

c.      Line 195: Consistently based on what? No effects of day.

d.      Line 195-196: Preference or an artefact of social dynamics? (see comment 10 c i)

e.      Line 198-202: Could the brush be mimicking maternal grooming?

f.       Line 208: There are studies in other species (for example see review by van de Weerd and Day, 2009. AABS, 116: 1-20.) that suggest that increasing the number of enrichment options is not as important as ensuring the enrichment sources remain novel.

g.      Line 209: Here observation period is referring to day not hour! I think you need to be explicit about which observation period you are referring to in your results and discussion.

h.      Line 212-214: I think this is a bit of a stretch. As previously discussed, the animals you have studied are young and there may be a functional reason why they are not engaging in play. Further, the enrichments were not frequently used and again, you cannot be sure that their use actually improves the welfare of these animals. Finally, you have low animal numbers, and your observations are on 11 day old calves in pairs in a particularly housing environment, so I recommend caution in extrapolating what you have found to any other environment. You either need to better develop this argument with reliance on the literature and consideration of the function of the enrichments, or remove this from the discussion.

i.        Lines 215-221: Again, this could be indicative of social relationships. Also, could also related to nutritional status rather than preference. What I am saying is that preference is just one of a few explanations of these results, and the other possibilities deserve consideration.

j.        Line 247-251: And a consideration of the nature and purpose of the enrichment!

k.      Line 256-257: Remove “Work in a controlled environment is needed to discern if calves are motivated to use brushes on their head regardless of fly load”. I don’t think this is relevant to your specific results.

9.      Figures

a.      Figure 2

                                                              i.     Brush and rope use in s/h would be easier to interpret as mins/hr

                                                            ii.     Insert “,” after “7 calves”

                                                          iii.     Would be useful to have a reference line on the charts indicating the time they were fed

b.      Figure 3

                                                              i.     Is there an outlier pulling the last value on Figure 3a up?

c.      Figure 4

                                                              i.     Brush use/rope use ratio be related to social effects, rather than simple preference. For example, if the animal with greater resource holding potential in the pair (larger, a little older, different temperament) is occupying one resource then the options of the second are limited. This needs to be discussed.

Author Response

We thank the reviewer for their comments and positive feedback. We have attended to each point individually in the PDF enclosed.
